# Asymmetric contextual effects in age perception

Deema Awad[1], Colin W. G. Clifford[2], David White[2] and Isabelle Mareschal[1]

[1]Department of Biological and Experimental Psychology, School of Biological and Chemical Sciences, Queen Mary University of London, London E1 4NS, UK
[2]The School of Psychology, UNSW Sydney, Sydney, New South Wales 2052, Australia

 

DA, 0000-0003-0487-3565; CWGC, 0000-0003-1043-6118;
DW, 0000-0002-6366-2699; IM, 0000-0003-4378-600X

**Subject Areas:**
psychology

**Keywords:**
age, face perception, flankers, contextual effects, contextual modulation

**Author for correspondence:**
Deema Awad
e-mail: d.awad@qmul.ac.uk

Perception is context dependent. For example, the perceived orientation of a bar changes depending on the presence of oriented bars around it. Contextual effects have also been demonstrated for more complex judgements, such as facial attractiveness or expression, although it remains unclear how these contextual facial effects depend on the types of faces surrounding the target face. To examine this, we measured the perceived age (a quantifiable measure) of a target face in the presence of differently aged faces in the surround. Using a unique database of standardized passport photos, participants were asked to estimate the age of a target face which was viewed either on its own or surrounded by two different identity flanker faces. The flanker faces were either both younger or both older than the target face, with different age offsets between flankers and targets of ±5, ±10, ±15, ±20 years. We find that when a target face is surrounded by younger faces, it systematically appears younger than when viewed on its own, and when it is surrounded by older faces, it systematically appears older than when viewed on its own. Surprisingly, we find that the magnitude of the flanker effects on perceived age of the target is asymmetric with younger flankers having a greater influence than older flankers, a result that may reflect the participants' own-age bias, since all participants were young. This result holds irrespective of gender or race of the faces and is consistent with averaging.

# 1. Introduction

## 1.1. Facial age estimation

How old we think someone is determines how we interact with them [1–3], and most people regularly make age judgements based on facial appearance. For example, when assessing the age of a suspect in crime scenes, or in daily tasks such as determining eligibility to buy alcohol or tobacco (for meta-analysis on age

perception, see [3]). Given the importance of making accurate facial age estimations, and how frequently we make them, one might assume our perceptual system is precise in judging age. Indeed, early research claimed that people make reliable age estimates, with errors of ±3–4 years [4,5]. However, those studies were limited in the number and quality of their stimuli. Faces change as we grow older, and ageing of facial skin is driven by factors such as gender, genetics, lifestyle, diet, smoking, and consuming drugs and alcohol [6,7]. Therefore, given the large amount of variability in faces, using a small set of test faces potentially biases age estimates to the specific test identities used. More recent work has addressed this issue by using both a larger number of stimuli and a broader sample of participants. In this case, the authors find that estimating age is less reliable than originally proposed with a reported mean error magnitude between ±6 years [8,9] and ±8 years [10]. Taken together, this reveals that people make errors in age judgements even for single faces viewed individually.

## 1.2. Contextual effects

Scenes are usually complex and rich in information, and although we assume our perception of the visual world is unbiased and accurate, this is not always the case. For example, it is well established that our perception of a visual target can change based on what surrounds it, a class of phenomena known as contextual effects (e.g. [11,12]). Contextual effects have been demonstrated for a wide range of low-level features such as orientation [13], motion [14,15], contrast [16,17] and size [18]. For example, in the well-known Ebbinghaus illusion, the perceived size of a central target circle surrounded by smaller circles appears larger than the same circle surrounded by larger circles [18–20].

Recently, there has been an increased interest in how contextual effects might influence our perception of less simplistic (higher level) objects such as faces, since in the natural environment we frequently encounter faces that vary systematically in appearance according to group classification (e.g. different age, race or gender groups). Therefore, to understand how we perceive individual faces we must understand how the presence of other faces may affect our perception, and contextual effects have been demonstrated for a range of other facial characteristics such as inference of emotion [21], perception of gender [21] identity [22], or facial attractiveness [23]. For example, Walker & Vul [23] presented participants with photographs of women viewed either in a group or in isolation and found that participants judged the same woman to be significantly more attractive when she was viewed within a group than when she was viewed alone. They concluded that this was the result of the visual system averaging visual information (about the faces), resulting in a prototypical (more attractive) face. However, in their study, participants never rated the attractiveness of the surrounding faces so it is unclear how their result may depend on the perceived attractiveness of the surrounding faces in the group.

Here, we were interested in investigating facial contextual effects and used age as our measure since, unlike facial expression or attractiveness, this feature has a quantifiable ground-truth. Based on previous experiments using faces reviewed above, we expected that contextual effects on the perception of a target face would be assimilative as a result of averaging. In this case, the target appears more similar to the context than it physically is [24] and these effects increase with increasing similarity between the target and flankers [24,25].

## 1.3. Own-group biases

There are also well-documented own-group biases that can influence our perception of a face. In the own-group bias, participants are better at making judgements about faces that are similar to themselves, such as the own-race bias [26], own-gender bias [27], own-species bias [28] and own-age bias [29]. These biases have been suggested to result from our greater perceptual experience with faces that match our own demographic profile. For example, Thornton et al. [30] showed that participants can make judgements about which race is most prominent in a group of faces, but critically, that those estimates are affected by participants' own race. This suggests that people can estimate the mean racial composition of a group, but that this estimate is biased by the observers' own characteristics. In that study, Asian and Caucasian participants were asked to make judgements about the racial composition of groups of faces that contained both races. They found a consistent effect of participants' own race on performance. When looking at the same stimulus, Asian and Caucasian participants made different judgements about the racial composition of a group of faces. Specifically, other-race faces were given more weight than own-race faces when assessing the composition of groups of faces. Another study by Kramer et al. [31] investigated whether social categorization (such as race or gender of participants and targets) would influence assimilation in ratings of facial attractiveness in serial dependency (or as authors call it 'sequential effects'), and their findings

suggest that own-race faces showed less opposite-sex assimilation, while other-race faces showed equal assimilation for opposite and same sex targets (i.e. female and male faces were not differentiated). Furthermore, they also found that for trials that varied in both race and sex, same category assimilation was significantly greater than other-category assimilation. However, it remains unclear how individual differences such as race or gender might influence spatial contextual effects of other facial characteristics such as age.

## 1.4. This study

Previous research has shown strong contextual effects on perception, suggesting that a high similarity between the target and the flankers increases the likelihood of assimilation. Here, we examined how the presence of a group of flanker faces influences people's judgements of a target face's age and whether this depended on the age difference (i.e. similarity) between the target and flankers. In experiment I, we measured age judgements of the target face when it was viewed on its own or flanked by younger/ older faces. We predicted that decreasing the age difference would increase assimilation effects, making the target appear more similar in age to the flankers' age. We also examined whether own-group biases, such as race and gender, influenced contextual effects of age perception. To do this, we measured age estimates of Caucasian test faces in both Caucasian and non-Caucasian participants (experiment I) and age estimates of East Asian faces in both East Asian and non-East Asian participants (experiment II). In experiment III, we also examined whether there was an own-gender bias in age estimates, and how age estimates of a target may depend on the genders of the target and flankers, since previous studies report largest contextual effects when the items in the ensemble are physically similar to each other [24].

Overall, we expected that age estimation of a target face would be influenced by the presence of flanking faces. If the magnitude of contextual effects in age perception are modulated by the similarity between the target and the flankers as has been reported in serial dependency effects discussed above, we would expect that when the age offset between the target and flankers was small, the flankers would produce assimilative effects (the target would appear more similar in age to the age of the flankers) but that these assimilative effects would disappear as the target and flanker faces became more dissimilar in age. Given well-documented own-group biases, we also expected that these contextual effects would be modulated by the participants' race and gender. In order to investigate whether contextual effects are modulated by own-group biases we also varied the race of the stimuli (target and flankers). Finally, as it has been suggested that increasing the similarity between the target and flankers leads to stronger contextual effects, we also expected that contextual effects would decrease when the flankers and target were of different genders [24].

# 2. Methods

## 2.1. Participants

A different set of participants was recruited for each of the three experiments (table 1 for demographics). All participants had normal or corrected to normal vision. The experiments were approved by the Human Research Ethics Committee of UNSW Sydney, and participants gave written informed consent to take part in the study. Participants were all UNSW psychology undergraduate students and were recruited through UNSW SONA system (Research management system) and either received course credit or were financially compensated $20 per hour for their time. In UNSW, approximately 30% of the first-year psychology cohort are international students and the majority of those are from East Asia. Thus, in experiment II, East Asian participants were from China, Japan, Korea and Vietnam, but Australian participants with East Asian heritage were not included.

## 2.2. Stimuli and display

Stimuli came from a large database of passport photographs from Australian citizens who had agreed to have their photograph used in academic research. The images were 426 by 536 pixels and were of Caucasian faces in experiment I and III, and of ethnically East Asian faces in experiment II. Stimuli were presented on one of three different laptops (Asus, Lenovo and HPs) and observers used a chin rest. At the viewing distance of 57 cm (all monitors were approximately 15″ by 10″ with a resolution of 1920 by 1080), each image subtended approximately at $8° \times 12°$ of visual angle. In the flanked

(*a*)     (*b*)     (*c*)

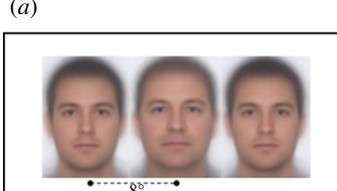     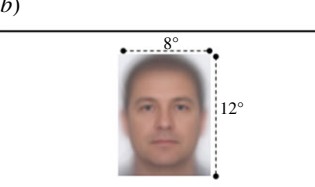     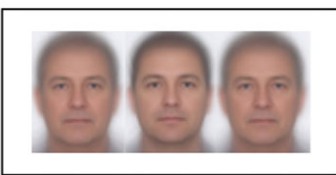

**Figure 1.** Illustration of the different flanker conditions. (*a*) Younger flankers' condition, (*b*) no flanker condition. (*c*) Older flankers' condition. To protect identities of the people contributing their passport images we are unable to publish individual images that were used in the study, images shown here are averages of 124 of the original faces used in the experiment and illustrate a ±15-year difference between flankers and target. In the experiment, the two flankers were always different identities.

**Table 1.** Summary of participant demographics and stimuli characteristics.

| exp | participants | flankers/target age offset | race of stimuli | flankers/target gender |
|---|---|---|---|---|
| I | $N = 68$ (25 males; 28 Caucasian, 40 other), age range = 17–35 years ($M = 19.77$; s.d. = 5.12). | ±5, ±10, ±15, ±20 | Caucasian | congruent |
| II | $N = 42$ (17 males; 16 East Asian, 26 other), age range = 17–29 years ($M = 19.13$; s.d. = 1.57). | ±5, ±10, ±15, ±20 | East Asian | congruent |
| III | $N = 60$ (23 males), age range = 17–25 years ($M = 19.10$; s.d. = 1.59). | ±5, ±15 | Caucasian | congruent/ incongruent |

conditions, the three images were presented horizontally with no gaps between them (approx. 8° centre to centre distance). Images with memorable facial characteristics such as scars or moles, open mouths or accessories were excluded. All faces were photographed forward facing with eyes angled directly at the camera with the same background colour. For security reasons and to protect the identity of the people contributing their passport images, we are unable to show the individual images used in the study. Figure 1 illustrates the general layout of the stimuli, using average faces that were created from 124 faces in our dataset covering a 4-year age range.

## 2.3. Pilot study

To select our test faces, we ran a pilot study in 68 observers (28 male, age range = 17–51, mean age = 20.46) to identify those faces whose perceived age most closely matched their veridical age (closest mean estimates and smallest standard deviation). The physical age of the images varied from 10 to 70 years old in steps of 5 years, and for each age step, we tested 20 different male identity and 20 different female identity faces. This ensured that in our main experiments, the flankers were likely to be perceived as younger/older than the target. Figure 2 plots the results of this pilot experiment (averaged across the 20 different identities for each target age step).

## 2.4. Procedure and design

In each of the experiments, participants were asked to give a two-digit numerical estimate of the perceived age of a target face by entering the number using a keyboard. Stimuli were presented in the centre of a laptop monitor and were presented against a homogeneous white background. There were three flanker conditions that were randomly interleaved: a no flankers condition (target viewed alone), a younger flankers condition and an older flankers condition. In the target flanked conditions, participants were informed that the target face was always in the middle surrounded (horizontally) by two different identity faces and that they should estimate the target face age only (figure 1). In the flanked conditions, the gender of the target and the flankers was always kept the same except for experiment III. For each age of target face tested, there were six different possible target identities (three male and three female). Each target face identity was shown once in each of the three flanker

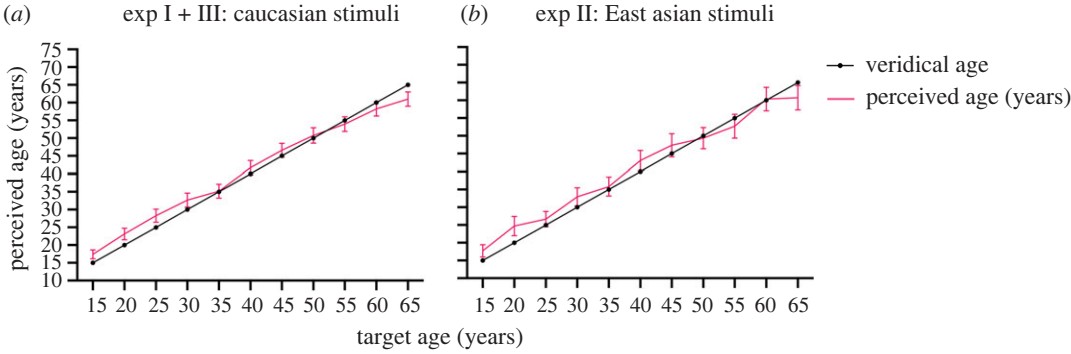

**Figure 2.** Age estimates of 20 female identity and 20 male identity faces for each target age from 15 to 65 in steps of 5 years when viewed alone for (*a*) Caucasian (experiment I + III) and (*b*) East Asian (experiment II) stimuli. The solid pink line illustrates the perceived age (participants' response) and the solid black line illustrates the veridical age of the faces. Error bars represent 95% confidence interval (CI).

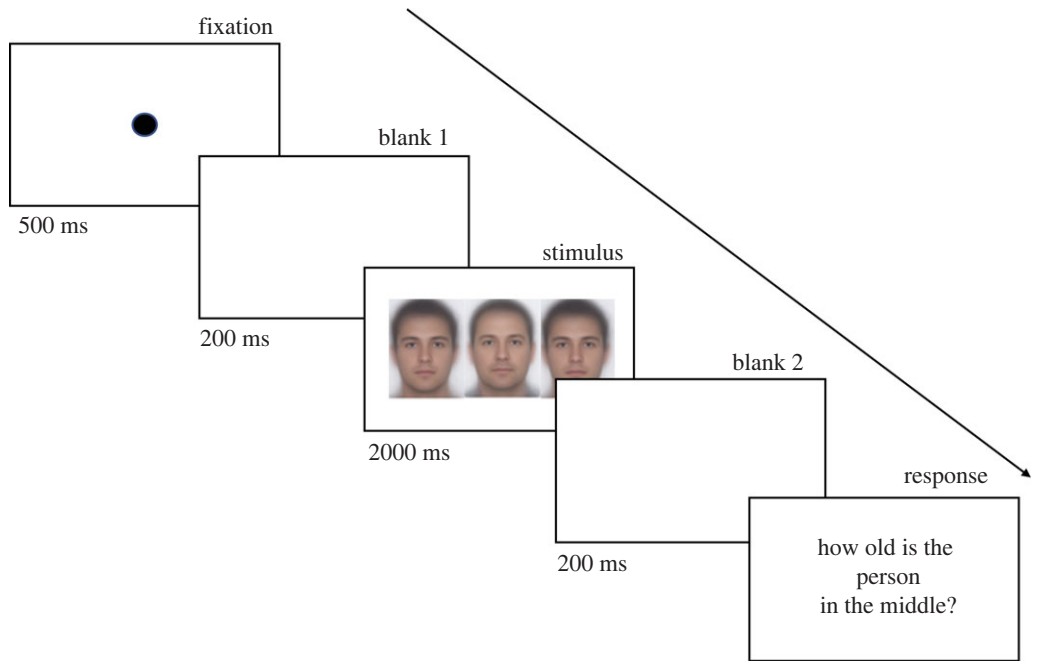

**Figure 3.** Timeline of a single trial, here showing stimulus illustrating (averaged faces) a flanked condition with 15 years younger flankers of a 40-year-old male target.

conditions, resulting in a total of 18 trials for each target age (six with younger flankers, six with older flankers and six with no flankers). Each trial began with a 500 ms fixation point, followed by a blank screen for 200 ms, followed by stimulus presentation for 2000 ms, followed by 200 ms blank screen, following which participants were instructed to type in a two digit response (between 01 and 99) and press 'enter' to move to the next trial. Figure 3 illustrates the sequence of a trial. Participants were not informed about the age range of the stimuli and no feedback was given. Stimulus presentation was randomized throughout the experiment and participants were free to take breaks during the experiment. The experiment took approximately 1 h to complete.

In all the experiments, we used a block design for stimulus presentation, such that observers were tested on a single magnitude of target–flanker age offset in a block. There were four blocks (with older and younger flankers randomly interleaved within the block), therefore, the age offset between the target and the flankers had one of eight possible values: ±5, ±10, ±15 and ±20 years. In Experiment III only, offsets tested were of ±5 and ±15 years. Since our dataset contained faces from people aged between 7 and 70 years of age, the range of target ages we were able to test depended on the age offset between flankers and target.

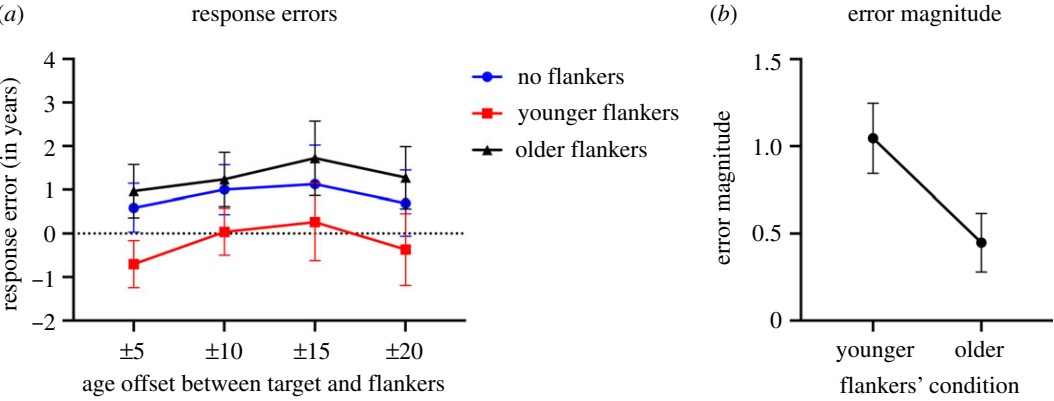

**Figure 4.** Experiment I: (*a*) Average response errors (calculated as the perceived age minus the veridical age) for the target faces when viewed with younger flankers (red lines), no flankers (blue lines) and older flankers (black lines) averaged across the different age targets viewed in each of the four target–flanker age offsets (±5, ±10, ±15, ±20). Error bars represent 95% confidence interval (CI). (*b*) Illustration of the response error magnitude caused by the presence of younger and older flankers. Error bars represent 95% confidence interval (CI).

## 2.5. Data analysis

In order to compare participants' age estimations across the different flanker conditions (no flankers, younger flankers, older flankers), the data from separate target ages (averaged across participants in each target–flanker age offset) were combined separately for each flanker condition. If the presence of flankers had no effect on the perceived age of the target then we would expect age estimates to be similar across the three conditions, while target age estimates that deviate systematically away from those in the unflanked condition would suggest contextual effects.

# 3. Results

## 3.1. Experiment I

In this experiment participants provided age estimates of Caucasian target faces viewed on their own or surrounded by two different identity flankers of the same gender and race. The flanker faces were either both younger or both older than the target face, with different age offsets between flankers and targets of ±5, ±10, ±15 and ±20 years. In order to examine the influence of the flankers we calculated the difference between the veridical age and the estimated age (termed response error) for the three flanker conditions averaged errors across all target ages for each age offset. Figure 4*a* plots these averaged response errors as a function of the different target flanker age offset conditions. In order to examine the effects of flankers on the target age estimation, we performed a two-way repeated measures ANOVA (Age offset (four levels) X Flanker condition (three levels)). Mauchly's test indicated that the assumption of sphericity had been violated for Flanker condition and for Age offset, therefore Greenhouse–Geisser corrections are reported. We found a significant main effect of Flanker condition $F_{1.51,99.60} = 97.47$, $p < 0.001$, $\eta_p^2 = 0.60$. However, we do not find a significant main effect of Age offset $F_{2.06,136.30} = 1.13$, $p > 0.050$, $\eta_p^2 = 0.08$. Furthermore, there was no significant interaction between Age offset and Flanker condition $F_{4.3,283.77} = 1.09$, $p > 0.050$. To further examine the main effect of Flanker condition, we ran a series of paired *t*-tests (combined across the different age offsets since there was no interaction). After Bonferroni correction, we found a significant difference $t_{67} = 10.29$, $p < 0.001$ in the response errors between the no flankers condition ($M = +0.85$, s.d. = 1.83) and the younger flankers condition ($M = -1.92$, s.d. = 1.66). We also found a significant difference $t_{67} = -5.19$, $p < 0.001$ between the no flankers condition ($M = +0.85$, s.d. = 1.83) and the older flankers condition ($M = 1.30$, s.d. = 1.96), and a significant difference $t_{67} = -10.92$, $p < 0.001$ between the younger flankers ($M = -1.92$, s.d. = 1.66) condition and the older flankers condition ($M = 1.30$, s.d. = 1.96).

Lastly, in order to compare the magnitude of the response error elicited by the younger and older flankers we calculated the absolute error magnitude (No flankers–Younger flankers) and (No flankers–Older flankers). For simplicity, and since there was no interaction between flankers' conditions and age offsets, we combined the data for all the age offsets together and ran a one paired sample *t*-test

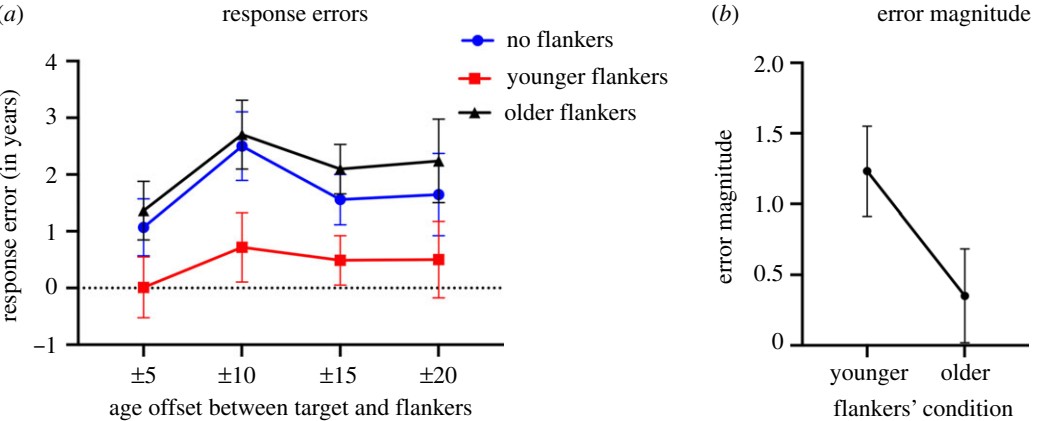

**Figure 5.** Experiment II: (*a*) Average response errors (calculated as the participant response minus the target veridical age) for target faces when viewed with younger flankers (red lines), no flankers (blue lines) and older flankers (black lines) averaged across the different age targets viewed with each of the four target–flanker age offsets (±5, ±10, ±15, ±20). Error bars represent 95% confidence interval (CI). (*b*) Illustration of the response error magnitude caused by the presence of younger and older flankers. Error bars represent 95% confidence interval (CI).

which revealed a significant difference between the two $t_{67} = 4.69$, $p < 0.001$, indicating that the younger flankers ($M = 1.04$, s.d. = 0.83) resulted in a larger response error than the older flankers ($M = 0.44$, s.d. = 0.69).

## 3.2. Experiment II

In a different set of participants, we repeated Experiment I but using East Asian faces as targets and flankers. We sought to investigate whether the effects in the previous experiment were limited to Caucasian test faces, and whether the contextual effects were modulated by participants' own-group biases. Figure 5*a* plots the averaged response errors in the different target flanker offset conditions. As in experiment I, we performed a two-way repeated measures ANOVA (Age offsets (four levels) X Flanker conditions (three levels)). Mauchly's test indicated that the assumption of sphericity had been violated for Flanker conditions and for Age offsets, therefore Greenhouse–Geisser corrections are reported. We found a significant main effect of Flanker conditions $F_{1.33,43.85} = 32.41$, $p < 0.001$, $\eta_p^2 = 0.50$. However, we did not find a significant effect of age offset $F_{2.32,76.41} = 1.07$, $p > 0.050$, nor a significant interaction between Age offsets and Flanker condition $F_{3.02,99.55} = 0.77$, $p > 0.050$. To further examine the main effect of Flanker condition, we ran a series of paired $t$-tests (combined across the different age offsets), and after Bonferroni correction, we found a significant difference $t_{41} = 7.88$, $p < 0.001$ in the response errors between the no flankers condition ($M = +1.64$, s.d. = 2.20) and the younger flankers condition ($M = 0.44$, s.d. = 2.17). We also found a significant difference $t_{41} = -2.10$, $p < 0.050$ between the no flankers condition ($M = +1.64$, s.d. = 2.20) and the older flankers condition ($M = 2.00$, s.d. = 2.17). Finally, we also found a significant difference $t_{41} = -5.97$, $p < 0.001$ between the younger flankers condition ($M = 0.44$, s.d. = 2.17) and the older flankers condition ($M = 2.00$, s.d. = 2.17).

As in experiment I, we compared the magnitude of the response error for the younger and older flankers. A sample $t$-test on the error magnitude revealed a significant difference between the two conditions $t_{41} = 4.70$, $p < 0.001$, indicating that the younger flankers ($M = 1.23$, s.d. = 0.99) resulted in a larger response error than the older flankers ($M = 0.35$, s.d. = 1.03).

To investigate the influence of an own-race bias on contextual effects of facial age, we combined the response error data from experiment I (Caucasian faces) and experiment II (East Asian faces) and divided the participants from the two experiments into groups of stimulus–participant race congruency (i.e. Caucasian participants judging Caucasian faces) or stimulus–participant race incongruency (i.e. non-Caucasian participants judging Caucasian faces) in the two experiments. We then ran a repeated mixed ANOVA (Flanker condition' (three levels)) with participant–stimulus race congruence as a between-subjects factor. Since we did not find an interaction between target–flanker age offsets and flanker condition, we combined the different age offsets. Figure 6 plots the averaged response errors for each flanker condition for each participant group (congruent and incongruent). We find a main effect of flanker condition $F_{1.40,144.66} = 92.97$, $p < 0.001$, $\eta_p^2 = 0.474$. However, we do not find a

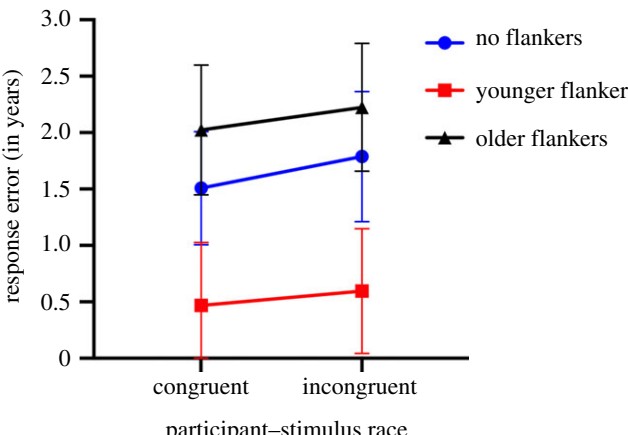

**Figure 6.** Illustration of the averaged response error for the different flanked conditions as a function of the participant–stimulus race congruency. Error bars represent 95% confidence interval (CI).

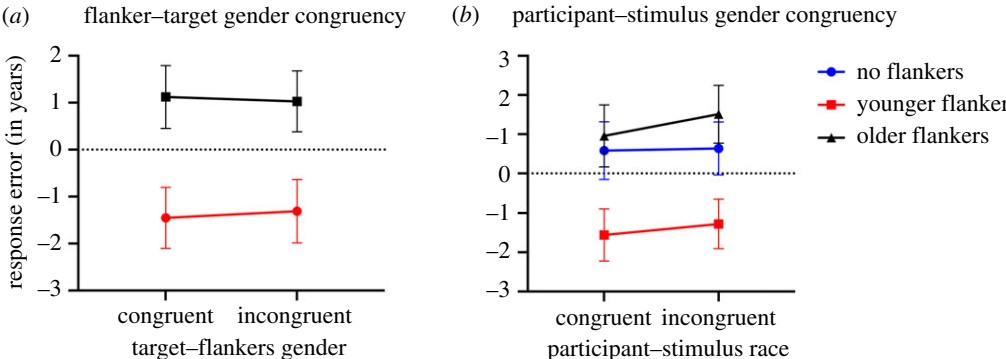

**Figure 7.** Experiment III: (*a*) Average response errors of targets in trials with congruent and incongruent gender for target and flankers for both younger flankers (red lines) and older flankers (black lines). (*b*) Average response errors based on participants-stimulus gender congruency for younger flankers, older flankers and no flankers. Error bars represent 95% confidence interval (CI).

significant interaction between race congruency and flanker conditions $F_{1.40,144.66} = 0.203$, $p > 0.050$, suggesting that contextual effects on facial age were not modulated by an own-race bias.

## 3.3. Experiment III

In this experiment, we investigated whether gender influenced age contextual effects. Figure 7*a* plots the response errors in the different flanked conditions separately for the flankers–target gender congruent and incongruent trials. Figure 7*b* plots these data based on congruency of participant and target gender (reported by passport data for stimuli and self-reported for participants). In order to examine the effects of participant–stimulus gender congruency on facial age ensemble representation, we ran a two-way repeated measures ANOVA (participant–stimulus gender congruency (two levels) X Flanker condition (three levels)). Since in previous experiments, we did not find a main effect of age offsets, or an interaction between age offsets and flanker condition, we combined the two different age offsets together. Mauchly's test indicated that the assumption of sphericity had been violated therefore Greenhouse–Geisser corrections are reported. Consistent with the previous experiments, we find a significant main effect of Flanker condition $F_{1.43,83.04} = 48.49$, $p < 0.001$, $\eta_p^2 = 0.455$, but no effect of participant–stimulus gender congruency $F_{1,59} = 2.39$, $p > 0.050$, $\eta_p^2 = 0.040$, and no significant interaction between flanker conditions and participant–stimulus gender congruency $F_{2,116} = 1.27$, $p > 0.050$, $\eta_p^2 = 0.021$.

In order to examine whether gender congruency between the flankers and targets affected the flankers' influence on age perception of the target, we compared response error for the congruent and incongruent conditions for each of the two flanker conditions. We ran a two-way repeated measures ANOVA (Flanker condition (two levels) X Flankers–Target gender congruency (two levels)). We found a significant main effect of Flanker condition $F_{1,59} = 60.12$, $p < 0.001$, $\eta_p^2 = 0.51$. However, we did not

find a significant effect of Flankers–Target gender congruency $F_{1,59} = 0.05$, $p > 0.050$, $\eta_p^2 = 0.001$, nor did we find a significant interaction between Flanker conditions and Flankers–Target gender congruency $F_{1,59} = 1.59$, $p > 0.050$, $\eta_p^2 = 0.000$.

# 4. General discussion

We examined how the presence of flanking faces affected the perceived age of a target face in three different experiments. Given the evidence for increased assimilation with higher similarity between target and flankers (e.g. gender [32], identity [33] and attractiveness [31,34]), we also examined whether the flanking contextual effects were influenced by the age similarity between the flankers and the target. Finally, we examined whether these effects were modulated by either the stimulus (age, gender and race) or participant characteristics (gender or race).

We find evidence for contextual effects on age perception, which is consistent with earlier findings on other facial contextual effects. We find that the perceived age of a face changes when it is viewed surrounded by other faces compared with when it is viewed on its own. Specifically, we find a systematic bias for faces to appear younger when they are flanked by younger faces compared with when flanked by older faces or viewed on their own. We also find that faces appear older when they are flanked by older faces compared with when flanked by younger faces or viewed on their own. Surprisingly, the magnitude of the error in perceived age of the target towards that of the flanker was larger in the presence of younger flankers than in the presence of older flankers. It is also worth noting that in all three experiments when the targets were viewed on their own, participants *overestimated* the age of the targets (apart from the older targets, a result that is consistent with previous findings of [10]). Thus, our data show that the presence of flankers, both younger and older, bias participants' judgement of a target face's age.

What might underlie these changes in perceived age when the target face is flanked by other faces? It is possible that because of perceptual averaging of the faces, individual facial characteristics (such as wrinkles or blemishes) are smoothed out, making the faces in general appear younger. However, this would suggest that target faces should also appear younger in the presence of older flankers compared with when viewed alone, which is inconsistent with our results. One explanation is that there may be two phenomena at play that act in the same direction in the case of younger flankers, but in opposite directions when the flankers are older. Firstly, the perceptual averaging of the flankers and the target biases the target face to be perceived as younger (due to the smoothing of the faces). Secondly, an assimilative effect where the age of the target face appears similar to the age of the flankers. In the case of older flankers, if the two effects largely balance each other out this might explain the smaller response error caused by the assimilation towards the older flankers.

In addition, the effects we find with younger flankers might be reinforced if younger flankers are given more weight in the averaging calculation compared with older flankers, producing a larger shift in the perceived age of the target when surrounded by younger flankers only. This effect of the younger flankers on the perceived age of the target face is evident regardless of the age of the target faces, or the absolute age of the flankers (as long as they are younger than the target). It is also possible that this effect is influenced by the participants' own age. Across all experiments, the participants were young (approx. 20 years old) so it may be that there was an own-age bias at play, whereby they attended more to the younger flankers than the older ones, leading to greater influence of the younger faces in the average. This interpretation is consistent with previous results suggesting that attention may bias estimates of ensemble perception [35,36]. For example, Chong & Treisman [35] used a dual-task paradigm to vary the spatial deployment of attention. Participants made either estimates of the average size of a set of circles, or estimates of the size of a specific circle while also performing a concurrent unrelated task. They found that extracting the mean size was easier when performing a concurrent task that required global attention than when the task required focused attention and concluded that the accuracy of average size estimates is influenced by the distribution of attention. Similarly, McNair *et al.* [37] found that limiting attentional resources biased ensemble expression perception by limiting the number of faces used to estimate the average expression. Thus, if participants here were attending more to younger faces this might explain the increased error magnitude caused by the younger faces compared with older faces. In age estimations, studies have demonstrated participants' tendency to be faster and more accurate in recognizing faces of their own age than another age group [38] and their tendency to look longer at images of faces within their own age group [39,40]. For example, He *et al.* [39] examined younger and older adults' visual scan patterns as they passively viewed younger and older neutral faces. Their findings provided evidence for an own-age bias in visual inspection of younger and older faces and that those biases are largely due to

greater personal and social relevance of own-age rather than other-age faces, and due to the existence of more accessible and elaborated schemas for own-age than other-age faces. However, it is important to note that none of the experiments above examined visual inspection time of older and younger faces presented simultaneously nor how viewing time would influence contextual effects on facial age perception. Our data suggest that participants gave more weight to the younger faces in a group; however, we do not know how they looked at the different faces when they performed this task.

We were also interested in determining whether the magnitude of the age offset between the target and the flankers' ages influenced contextual effects on age perception. Clifford et al. [10] reported assimilative serial dependency whereby the perceived age of the target face was biased towards the age of a previously viewed face. However, adaptation studies by Schweinberger et al. [41] and O'Neil & Webster [42] demonstrated contrastive effects whereby the perceived age of a target face was shifted away from the age of the adaptor face. We find here that when the target and flanker faces are viewed simultaneously, only assimilative effects occur, regardless of the age difference/similarity between the flankers and target. Notably, even though assimilation occurs, its effects on perceived age estimates are subtle (response errors of no more than 2 years on average). On the one hand, subtle effects are reassuring because this suggests that participants were not reporting the age of the flankers instead of the target, since if this were the case, the target age estimate would be much closer to that of the flankers. On the other hand, since the shifts in age estimations are subtle (2–3 years shifts on average) this might explain why those effects were not significantly influenced by the different factors such as stimuli/participants' gender or race.

In experiments I and II, we also investigated whether contextual effects on age perception are modulated by race similarity between the participants and stimuli. Research on person perception has shown an own-race bias where people tend to have better recall or recognition for faces of the same race as them (for a review, see [26]), which may simply reflect an exposure effect [43–45]. For example, Thornton et al. [46] presented Asian and Caucasian participants with brief displays containing 16 faces and asked them to judge whether there were more Asian faces or Caucasian faces present. Asian participants appeared to weight other-race faces more heavily than Caucasian participants. However, in the above study the race of the stimuli was task-relevant, while here it was task-irrelevant. Additionally, in Thornton et al.'s experiment the stimuli were of mixed races while here the targets and flankers were of the same race. Our data suggest that regardless of the race of the stimuli being tested or the race of the participants, facial age estimates shifted towards the age of the flankers across all conditions.

Finally, our data show no evidence of an own-gender bias for age perception. Schweinberger et al. [41] found that after adaptation to older faces, test faces were classified as younger than when the adaptors were young faces. Importantly, their experiment showed the same effect (although reduced) when the gender of the adaptors and test faces were mismatched. Here, we investigated whether the effects we found were limited to similar genders between flankers and targets. Consistent with Schweinberger et al.'s [41] results, we find that mixing the gender of the target and flankers did not abolish contextual modulation of facial age; participants' age estimation of a target face shifted towards the flankers regardless of gender. However, contrary to Schweinberger et al. [41], we did not observe any reduction in the contextual effects when the gender of flankers/target differed.

Ethics. The experimental procedures in this study were approved by UNSW Human Research Ethics Committee, which adheres to the Declaration of Helsinki. All participants gave informed consent.

Data accessibility. Data available from the Dryad Digital Repository: https://doi.org/10.5061/dryad.7wm37pvr2. To protect the identity of the people contributing their passport image for use in the study, we are unable to make available individual stimulus images.

Authors' contribution. D.W. supplied stimulus materials. D.A. implemented the experiments, collected data and ran analysis under the supervision of C.W.G.C. and I.M. All authors approved the final version of the manuscript.

Competing interests. We have no competing interests.

Funding. D.A. was funded by the Economic and Social Research council, C.W.G.C. was funded by Australian Research Council Discovery Project DP190100491, D.W. was funded by Australian Research Council Linkage Project LP160101523, and I.M. was funded by the Medical Research Council (grant no. MR/S011307/1), and the Waterloo Foundation (grant no. 2072-3513).

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
