## [Reviewer comments · Royal Society Open Science]

Review History

RSOS-200936.R0 (Original submission)

Review form: Reviewer 1

Is the manuscript scientifically sound in its present form?

No

Are the interpretations and conclusions justified by the results?

No

Is the language acceptable?

No

Do you have any ethical concerns with this paper?

No

Have you any concerns about statistical analyses in this paper?

Yes

Recommendation?

Reject

Comments to the Author(s)

Thank you for the opportunity to review the manuscript, "Asymmetric contextual effects in age perception".

The ms presents a series of three experiments in which participants were asked to judge the age of a target face that was flanked by two non-target flanker faces. The flanker faces differed in age from the target face by ± 5 , ± 10 , ± 15 , ± 20 . Experiments 1 & 3 tested Caucasians on Caucasian faces. Experiment 2 tested East Asians on East Asian faces (at least according to Table 1, although the data suggest otherwise?). In the first two experiments, the flanker and target faces were the same gender. In the third experiment, the gender of the flanker faces was manipulated relative to the target face, so that there was a same gender and a different gender condition. The key finding was that faces were judged as younger when they were flanked with younger faces. There was also a trend for faces to be judged as older when they were flanked with older faces, but this effect was weaker and non-significant.

Understanding how context influences perceptual judgements is an important task for the literature, and overall this work was empirically sound. However, I think that considerably more work is needed to make the most of the raw material presented in this ms.

INTRODUCTION

– A more thorough review of the literature would strengthen this ms. I think the themes the introduction covers are the right ones, but the literature needs more rigorous analysis (e.g., on age judgements). The introduction should also give more consideration to why assimilative vs contrastive contextual effects occur, in a way that clearly leads to the hypotheses. In the present version, the link between the introductory literature review and the hypotheses is unclear (e.g., not clear from literature review why "we would expect that when the age offset between the target and flankers was small, the flankers would produce assimilative effects", p10 ln33). Serial dependency and adaptation need unpacking.

METHODS

– Some relevant details are missing. How much were participants compensated? How long did the experiment take? Why were images with memorable characteristics excluded rather than edited? Were flankers two different identities? (I assume so, but the use of average faces in the figure is misleading in this way.)

– What criteria were used to select the stimuli based on the pilot study data? What were the characteristics of the final stimulus set? While this information can be gleaned from the supplemental materials, I think it is important that the stimuli be properly characterised in the main text.

RESULTS

– Overall, I think there are two key conclusions that can be soundly drawn from the data: (1) that the presence of flankers biases age judgements of a target face in the direction of the flankers; and (2) this effect is robust in that it is not impacted by whether observers and stimuli are the same race or not, nor by whether the gender of the flankers matches that of the target or not.

– I am however not convinced that the key effect is truly asymmetrical. Looking at the bottom right panel of Figures 3 and 4, I can see that the magnitude of the bias is larger for younger flankers than for older ones. However, it still looks to me like there is some bias towards older flankers too. I can't find the t-test results for the older flanker vs no flankers analyses though. I also find it interesting that, for the older flankers, the effect seems to be linear, getting larger with increasing offsets. This again suggests to me that something real and of interest is going on here. Statistically, to establish this effect is asymmetrical, a test that compares the bias for younger flankers to that for older flankers is needed. Furthermore, even if this test revealed a significant difference, the fact that only young adults were tested suggests that this asymmetry could be driven by some own-age bias. If there is strong evidence of asymmetry after further interrogation, I would like to see the title and abstract reflect that this is in young adults only, to protect against the finding from being interpreted more broadly.

MINOR

– $p=.000$ should be reported as $p<.001$

Review form: Reviewer 2

Is the manuscript scientifically sound in its present form?

Yes

Are the interpretations and conclusions justified by the results?

Yes

Is the language acceptable?

Yes

Do you have any ethical concerns with this paper?

No

Have you any concerns about statistical analyses in this paper?

No

Recommendation?

Accept with minor revision (please list in comments)

Comments to the Author(s)

In this study, the authors investigated whether flanking faces can influence age ratings of a central face. They found that flankers with younger identities lead to younger ratings on the central identity.

The study is well conducted, the data are solid and the manuscript is well written with several references. The effect itself is interesting and new in the literature. I have only a few minor comments, which mostly aim at improving the paper.

Several confounding factors are examined, but there is a missing one. The age of the observers (who are all relatively young) could influence the age ratings and, thus, be responsible for the reported effect. To be fair, the authors mention this possible confound in the Discussion, but I think that section should be at least expanded. For example, does the effect correlated with the observers' age? Alternatively, are there any age effects on the single face ratings? If yes, that could ever further support the confound. More generally, it is important to acknowledge that, if the effect is due to the age of the observers, the proposed effect is less general than what it is discussed (but it is still important).

Minor minor comments:

1. It would be useful to have the legend IN the figure.
2. Page 8 lines 43-54: some citations should be provided in support of this claim.
3. Figure 1supp: "percieved"

Review form: Reviewer 3

Is the manuscript scientifically sound in its present form?

Yes

Are the interpretations and conclusions justified by the results?

No

Is the language acceptable?

Yes

Do you have any ethical concerns with this paper?

No

Have you any concerns about statistical analyses in this paper?

Yes

Recommendation?

Major revision is needed (please make suggestions in comments)

Comments to the Author(s)

See attached file (Appendix A).

Decision letter (RSOS-200936.R0)

Dear Miss Awad,

The editors assigned to your paper ("Asymmetric contextual effects in age perception") have now received comments from reviewers. We would like you to revise your paper in accordance with the referee and Associate Editor suggestions which can be found below (not including confidential reports to the Editor). Please note this decision does not guarantee eventual acceptance.

Please submit a copy of your revised paper before 12-Aug-2020. Please note that the revision deadline will expire at 00.00am on this date. If we do not hear from you within this time then it will be assumed that the paper has been withdrawn. In exceptional circumstances, extensions may be possible if agreed with the Editorial Office in advance. We do not allow multiple rounds of revision so we urge you to make every effort to fully address all of the comments at this stage. If deemed necessary by the Editors, your manuscript will be sent back to one or more of the original reviewers for assessment. If the original reviewers are not available, we may invite new reviewers.

- Data accessibility

If you wish to submit your supporting data or code to Dryad (<http://datadryad.org/>), or modify your current submission to dryad, please use the following link:
<http://datadryad.org/submit?journalID=RSOS&manu=RSOS-200936>

- Competing interests

- Authors' contributions

- Acknowledgements

- Funding statement

on behalf of Dr Bruno Rossion (Associate Editor) and Essi Viding (Subject Editor)
openscience@royalsociety.org

Comments to Author:

Reviewers' Comments to Author:

Reviewer: 1

Comments to the Author(s)

Thank you for the opportunity to review the manuscript, "Asymmetric contextual effects in age perception".

The ms presents a series of three experiments in which participants were asked to judge the age of a target face that was flanked by two non-target flanker faces. The flanker faces differed in age from the target face by ± 5 , ± 10 , ± 15 , ± 20 . Experiments 1 & 3 tested Caucasians on Caucasian faces. Experiment 2 tested East Asians on East Asian faces (at least according to Table 1, although the data suggest otherwise?). In the first two experiments, the flanker and target faces were the same gender. In the third experiment, the gender of the flanker faces was manipulated relative to the target face, so that there was a same gender and a different gender condition. The key finding was that faces were judged as younger when they were flanked with younger faces. There was also a trend for faces to be judged as older when they were flanked with older faces, but this effect was weaker and non-significant.

Understanding how context influences perceptual judgements is an important task for the literature, and overall this work was empirically sound. However, I think that considerably more work is needed to make the most of the raw material presented in this ms.

INTRODUCTION

– A more thorough review of the literature would strengthen this ms. I think the themes the introduction covers are the right ones, but the literature needs more rigorous analysis (e.g., on age judgements). The introduction should also give more consideration to why assimilative vs contrastive contextual effects occur, in a way that clearly leads to the hypotheses. In the present version, the link between the introductory literature review and the hypotheses is unclear (e.g., not clear from literature review why "we would expect that when the age offset between the target and flankers was small, the flankers would produce assimilative effects", p10 ln33). Serial dependency and adaptation need unpacking.

METHODS

– Some relevant details are missing. How much were participants compensated? How long did the experiment take? Why were images with memorable characteristics excluded rather than edited? Were flankers two different identities? (I assume so, but the use of average faces in the figure is misleading in this way.)

– What criteria were used to select the stimuli based on the pilot study data? What were the characteristics of the final stimulus set? While this information can be gleaned from the supplemental materials, I think it is important that the stimuli be properly characterised in the main text.

RESULTS

– Overall, I think there are two key conclusions that can be soundly drawn from the data: (1) that the presence of flankers biases age judgements of a target face in the direction of the flankers; and (2) this effect is robust in that it is not impacted by whether observers and stimuli are the same race or not, nor by whether the gender of the flankers matches that of the target or not.

– I am however not convinced that the key effect is truly asymmetrical. Looking at the bottom right panel of Figures 3 and 4, I can see that the magnitude of the bias is larger for younger flankers than for older ones. However, it still looks to me like there is some bias towards older flankers too. I can't find the t-test results for the older flanker vs no flankers analyses though. I also find it interesting that, for the older flankers, the effect seems to be linear, getting larger with increasing offsets. This again suggests to me that something real and of interest is going on here. Statistically, to establish this effect is asymmetrical, a test that compares the bias for younger flankers to that for older flankers is needed. Furthermore, even if this test revealed a significant difference, the fact that only young adults were tested suggests that this asymmetry could be driven by some own-age bias. If there is strong evidence of asymmetry after further interrogation, I would like to see the title and abstract reflect that this is in young adults only, to protect against the finding from being interpreted more broadly.

MINOR

– $p=.000$ should be reported as $p<.001$

Reviewer: 2

Comments to the Author(s)

In this study, the authors investigated whether flanking faces can influence age ratings of a central face. They found that flankers with younger identities lead to younger ratings on the central identity.

The study is well conducted, the data are solid and the manuscript is well written with several references. The effect itself is interesting and new in the literature. I have only a few minor comments, which mostly aim at improving the paper.

Several confounding factors are examined, but there is a missing one. The age of the observers (who are all relatively young) could influence the age ratings and, thus, be responsible for the reported effect. To be fair, the authors mention this possible confound in the Discussion, but I think that section should be at least expanded. For example, does the effect correlated with the observers' age? Alternatively, are there any age effects on the single face ratings? If yes, that could ever further support the confound. More generally, it is important to acknowledge that, if the effect is due to the age of the observers, the proposed effect is less general than what it is discussed (but it is still important).

Minor minor comments:

1. It would be useful to have the legend IN the figure.
2. Page 8 lines 43-54: some citations should be provided in support of this claim.
3. Figure 1supp: "percieved"

Reviewer: 3

Comments to the Author(s)

See attached file

Author's Response to Decision Letter for (RSOS-200936.R0)

See Appendix B.

RSOS-200936.R1 (Revision)

Review form: Reviewer 1

Is the manuscript scientifically sound in its present form?

Yes

Are the interpretations and conclusions justified by the results?

Yes

Is the language acceptable?

Yes

Do you have any ethical concerns with this paper?

No

Have you any concerns about statistical analyses in this paper?

No

Recommendation?

Accept as is

Comments to the Author(s)

The authors have addressed my initial comments satisfactorily. I think the changes to the analyses have produced a much cleaner and more compelling set of findings, which are appropriately interpreted.

Decision letter (RSOS-200936.R1)

Dear Miss Awad,

It is a pleasure to accept your manuscript entitled "Asymmetric contextual effects in age perception" in its current form for publication in Royal Society Open Science. The comments of the reviewer(s) who reviewed your manuscript are included at the foot of this letter.

on behalf of Dr Bruno Rossion (Associate Editor) and Essi Viding (Subject Editor)
openscience@royalsociety.org

Reviewer comments to Author:
Reviewer: 1

Comments to the Author(s)
The authors have addressed my initial comments satisfactorily. I think the changes to the analyses have produced a much cleaner and more compelling set of findings, which are appropriately interpreted.

Appendix A

My general impression is that there's nothing here that will preclude publication, but I think it'll take quite a chunk of work to get there. The analysis of the results is the part that really needs some attention. My comments on the analysis are as follows

1) The dependent variable mainly used in the plots and analyses is the response (in years). The upshot of this is that the majority of the graphs show a nice linear slope. The trouble here is that this hides the results of interest (the pattern across flanker age offset). That's because the main effect of target age is much larger than anything else. Now, we're not interested in the effect of target age. All this tells us is that people can judge people's ages. It's a large effect – but you can tell this by the simple fact that it's not particularly hard to tell a 30 year old from a 50 year old. I think the dependent variable should be the response minus the target age (response error). That way, the reader would actually be able to tell what's happening in the graphs rather than everything being overwhelmed by something that's not of central interest to the paper.

2) The construction of the ANOVAs is strange. You've got this factor Age Offsets which just doesn't make sense. As far as I can tell, it averages +15/-15 flankers, and +10/-10 flankers and so on. Now, a reasonable prior expectation is that the pluses and minuses should do opposite things to one another, and this this should be modulated by their magnitudes. Any such effect is basically cancelled out by the grouping within this factor. I think it needs more thought.

3) My personal preference would be to rethink the whole analysis approach. I think you should ask much more targeted questions such as “is there a difference between +ve and -ve flankers?” and then design a statistic to address this. So, for example, you could just collapse across everything but flanker condition and compare these three (which of course, is exactly what the ANOVA does.) Then you could look at a plot of response error over flanker value – so running from -15 to +15 on the x-axis. Then you could look to see whether some reasonable curve could describe the data ... along with parameter confidence limits calculated through bootstrapping and so on. The whole approach would be much more targeted and would be much more in the vein of the sort of analysis that you find in the vision literature ... these are all techniques that your co-authors are very familiar with. At the moment, the whole analysis just comes across as a bit of a mess with a whole bunch of inscrutable interactions and various t-tests that are shoved into the supplementary materials and then largely ignored.

4) The age offset graphs are perhaps the most informative graphs in the paper. I don't understand why there are no error bars. Whilst we're on the subject, what do the error bars on the other graphs show? Standard errors? The t-test done on the age offset data – this can be done better. At the moment you're just doing (presumably) a t-test to look at the difference between two columns of 4 numbers each. So you're chucking away all that information about the variability within your participant pool. There are a bunch of approaches that you could take, your co-authors will have some ideas.

5) The idea that non-significance means no effect. This runs through the paper and is just plain wrong. If you want to make the assertion that there's no effect, then there are statistics that you can use – again, your co-authors will know. Otherwise, it needs to be expunged from the paper, which means changes in the abstract, results and discussion.

6) The complexity of the third experiment. There are 6 factors – target age, target gender, flanker condition, age offset, flanker congruence and participant gender. It's really complicated ... to my mind, overcomplicated. I think, and this goes back to the points I've made above, a more targeted analysis would really help with this experiment. Also, from my reading you're using participant gender rather than gender congruence – why? I think you should be using the same approach as in the previous experiment.

So the major issue I have with the paper is really with the entire conceptualisation of the analysis. I also have a bunch of minor issues, these are as follows

1) P4 “perception of emotion” - personally, I think you perceive expression and then infer emotion. Others think different.

2) P5 – contrastive serial dependence with facial expressions. Have a look at the Liberman et al (2018) paper entitled “Serial dependence promotes the stability of perceived emotional expression depending on face similarity”.

3) You motivate your race/gender questions with the other race literature. I don't think it hangs together very well. Making people of different races make judgements of race is a bit different to people of different races or genders making judgements about age. Seeing how that might be differentially moderated by flanker is even more of a stretch. I think you need more/better motivation from the ingroup/outgroup literature.

4) P8 SONA – what is this?

5) How did you set/maintain viewing distance?

6) Pilot study. Really, I'd have liked to have seen the pilot study included in the main paper. I'm not a big fan of dumping stuff in supp mats. My view is that if you need it for the paper, then it should be in the paper. I can well understand that you might choose to keep the pilot in supp mats but can you please add a box plot of response minus target age over target age so that I can see something of the way the response distributions change over target age.

7) P13 bottom. “such that observers were tested on a single target-flanker age offset in a block”. Actually these appear to be negative/positive flanker pairs. To my mind, there are 8 age offsets, not 4.

8) Results ... I've commented in detail above about the results and I'd hope to see substantial changes and an increase in clarity with these ... but just to point out some of the non-sig means no effect errors – p19, sentence above Figure 5. P22, final sentence before General Discussion.

9) P22. The first and penultimate sentences of the first paragraph basically say the same thing.

10) P23. “this shift of perceived age of the target towards the flankers' ages was only significant with younger flankers”. Firstly, I can't remember seeing a statistic that actually supports this. I may have missed it, but I have been through the paper twice – so either it's not there, or you've not done enough to draw it to your readers' attention. Secondly, you can't make anything of this unless you have a significant difference between the effect with younger flankers and the effect with older flankers. In other words, if A is significant and B is not, this does not necessarily imply a significant difference between A and B. Have a read of Nieuwenhuis et al (2011) Erroneous analyses of interactions in neuroscience: a problem of significance. *Nature Neuroscience*, 14(9), 1105-1107.

Appendix B

Response to the editor

We would like to thank the editors for dealing with our manuscript and encouraging a resubmission. We would also like to thank the reviewers for their constructive suggestions. Based on their comments we have revised our manuscript substantially. Specifically, we implemented the two main changes suggested by reviewer 3. Firstly, we re-analysed the data for all the experiments based on response errors (difference between perceived and veridical age) rather than raw estimates of age judgements. Secondly, we averaged target ages across the experiments which allowed us to streamline our analysis substantially. When we analyse the data following the two suggestions above, we now find a (smaller) effect of the older flankers in addition to the previously reported effect of younger flankers that we now discuss in the paper. This change has allowed us to reformat the manuscript such that there are no supplementary materials. Below we address this, and all other concerns raised by the 3 reviewers.

We thank you for dealing with this submission and feel that the paper has been improved by these helpful suggestions.

Best wishes,

Deema Awad, Colin W. G. Clifford, David White, & Isabelle Mareschal

Response to reviewers

We thank all the reviewers for taking the opportunity to review the manuscript “Asymmetric contextual effects in age perception” and for their constructive comments. Below we include a point by point reply to the issues raised by each reviewer.

Reviewer 1

INTRODUCTION:

A more thorough review of the literature would strengthen this ms. I think the themes the introduction covers are the right ones, but the literature needs more rigorous analysis (e.g., on age judgements). The introduction should also give more consideration to why assimilative vs contrastive contextual effects occur, in a way that clearly leads to the hypotheses. In the present version, the link between the introductory literature review and the hypotheses is unclear (e.g., not clear from literature review why “we would expect that when the age offset between the target and flankers was small, the flankers would produce assimilative effects”, p10 ln33). Serial dependency and adaptation need unpacking.

- We thank the reviewer for their constructive comments. We agree that the previous introduction did not provide a thorough discussion about age judgements in general, this was because we sought to focus on contextual effects with respect to age judgements rather than on age judgements per se, as this has been previously covered by some of the co-authors (see Clifford, Watson & David, 2018) and others (see Rhodes, 2009). However, we agree that further discussion would strengthen the manuscript and have now revised the manuscript to include a more detailed discussion of age judgements. Please see the revised changes on page 3 (Introduction) lines 63-69.
- In terms of assimilative vs contrastive contextual effects, based on previous literature we expected that when the difference between the flankers and target was small (e.g. when items are similar, at least with respect to the feature of interest here: age), this would lead to assimilative effects and that when the difference was larger (i.e. the faces are less similar) this would lead to smaller effects (or to contrastive effects in some studies). However, in the revised manuscript we have toned down the discussion of contrastive effects as it was mainly based on literature on low level contextual effects rather than on high level facial attributes. We have reframed the hypothesis to state that we expect the assimilative effects to be modulated by the similarity between the target and the flankers. Please see the revised changes on page 6 lines 140-147.

METHODS

Some relevant details are missing.

- We thank the reviewer for pointing out the missing details. We have clarified the following issues below in the revised manuscript.

How much were participants compensated?

- Participants were compensated \$20 per hour or received course credits. Please see the revised changes on page 8 line 178.

How long did the experiment take?

- Each experiment took approximately 1 hour (based on participants' speed in entering their responses as it was self-paced). Please see the revised changes on page 11 line 243.

Why were images with memorable characteristics excluded rather than edited?

- Images with memorable characteristics (such as accessories or obvious facial characteristics such as beards) were excluded in order to minimise participants' memory of their previous responses with the faces (each target face was presented 3 times). Fortunately, we had a large number of faces to choose from (Australian passport datasets) so we were able to remove these images without substantially reducing the pool of photos we could use.

Were flankers two different identities? (I assume so, but the use of average faces in the figure is misleading in this way.)

- The two flankers were of two different identities. We appreciate that the averaged faces in the figures does not reveal this, and so have clarified it in the figure legend and in the methods section. Please see the revised changes on page 10 line 231.

What criteria were used to select the stimuli based on the pilot study data?

- Based on our pilot study (N=68), we selected 3 female faces and 3 male faces to be used for each of the target ages tested. The images chosen were those that had been judged closest to the veridical age of the person and had the smallest standard deviation of age estimates (no more than 1-2 years standard deviation).

What were the characteristics of the final stimulus set? While this information can be gleaned from the supplemental materials, I think it is important that the stimuli be properly characterised in the main text.

- The final stimulus set included 3 identities for female targets, and 3 identities for male targets for each of the target ages for each of the experiments (see line 234). In experiment I and III those were of Caucasian faces, and in experiment II those were of East Asian faces (see lines 188-189). All the faces had direct gaze towards the camera, and had no accessories (glasses, scarves, etc.) (see lines 194-196). Based on the pilot experiment, we chose the faces with the perceived age that is closest to the veridical age (based on participants' age estimations mean), and those with the smallest standard deviation (see lines 209-216).

RESULTS

Overall, I think there are two key conclusions that can be soundly drawn from the data: (1) that the presence of flankers biases age judgements of a target face in the direction of the flankers; and (2) this effect is robust in that it is not impacted by whether observers and stimuli are the same race or not, nor by whether the gender of the flankers matches that of the target or not. I am however not convinced that the key effect is truly asymmetrical. Looking at the bottom right panel of Figures 3 and 4, I can see that the magnitude of the bias is larger for younger flankers than for older ones. However, it still looks to me like there is some bias towards older flankers too. I can't find the t-test results for the older flanker vs no flankers analyses though. I also find it interesting that, for the older flankers, the effect seems to be linear, getting larger with increasing offsets. This again suggests to me that something real and of interest is going on here. Statistically, to establish this effect is asymmetrical, a test that compares the bias for younger flankers to that for older flankers is needed. Furthermore, even if this test revealed a significant difference, the fact that only young adults were tested suggests that this asymmetry could be driven by some own-age bias. If there is strong evidence of asymmetry after further interrogation, I would like to see the title and abstract reflect that this is in young adults only, to protect against the finding from being interpreted more broadly.

- We thank the reviewer for their constructive comments. Based on those comments (along with reviewer 3 comments) we have redone the analysis in the manuscript. Previously, we ran separate analysis (t-tests) for each of the target ages because we were interested in whether certain target ages might be more/less biased by the flankers, and how this might interact with the participants' age. However, thinking about this more in the context of your query as well as that of reviewer 3, we have decided to look at overall flanker effects, regardless of the target age. To do this, we examined the response error (e.g. the difference between the estimated age and

veridical age) instead of the actual age given by participants, averaged across all target ages in each target-flanker offset condition. Please see the revised changes in the results section starting from page 12.

- The new analysis replicates our previous findings with younger flankers, but additionally it revealed a significant effect of the older flankers. We examined the size of this effect as suggested by the reviewer and report that the younger flankers lead to a significantly larger response error than the older flankers. Therefore, although both types of flankers bias perception of a target, the magnitude of this effect is asymmetrical. We have now revised the analysis and results sections to include this new analysis. We discuss, as suggested by the reviewer, that this might be due to participants' own-age biases in both the general discussion and the abstract. Please see this discussion changes from lines 423-456.

MINOR

p=.000 should be reported as p<.001

- Thank you, this has now been fixed.

Reviewer 2

Several confounding factors are examined, but there is a missing one. The age of the observers (who are all relatively young) could influence the age ratings and, thus, be responsible for the reported effect. To be fair, the authors mention this possible confound in the Discussion, but I think that section should be at least expanded. For example, does the effect correlated with the observers' age? Alternatively, are there any age effects on the single face ratings? If yes, that could ever further support the confound. More generally, it is important to acknowledge that, if the effect is due to the age of the observers, the proposed effect is less general than what it is discussed (but it is still important).

- We thank the reviewer for their constructive comments. It is true that we find a larger effect with the younger flankers. However, based on a suggestion from reviewer 3 we have decided to change the analysis to (a) ignore the target ages since we had no a priori expectation about target age and (b) to examine the data as a function of response error (participant's response minus the target veridical age) rather than on the raw age estimations. Based on the new analysis we replicate the effect of younger flankers, but we also now report a significant but smaller effect of older flankers. This new analysis is now in the revised results section of each experiment. Please see the revised changes in the results section starting from page 12.
- We agree that it is important to acknowledge that the participants' own age might play a substantial role in this effect and expand on this further in the abstract,

introduction, and in the discussion. Please see this discussion changes from lines 423-456.

Minor comments:

It would be useful to have the legend IN the figure.

- We have fixed this; all plots have legends in the figure in the revised manuscript.

Page 8 lines 43-54: some citations should be provided in support of this claim.

- Thank you, this has been fixed.

Figure 1supp: "percieved"

- Thank you, this has been fixed.

Reviewer 3

My general impression is that there's nothing here that will preclude publication, but I think it'll take quite a chunk of work to get there. The analysis of the results is the part that really needs some attention. My comments on the analysis are as follows

1) The dependent variable mainly used in the plots and analyses is the response (in years). The upshot of this is that the majority of the graphs show a nice linear slope. The trouble here is that this hides the results of interest (the pattern across flanker age offset). That's because the main effect of target age is much larger than anything else. Now, we're not interested in the effect of target age. All this tells us is that people can judge people's ages. It's a large effect – but you can tell this by the simple fact that it's not particularly hard to tell a 30 year old from a 50 year old. I think the dependent variable should be the response minus the target age (response error). That way, the reader would actually be able to tell what's happening in the graphs rather than everything being overwhelmed by something that's not of central interest to the paper.

- We thank the reviewer for this useful suggestion. We agree that an analysis that takes into account target age is less interesting and also more likely to camouflage more subtle effects due to the flankers. Originally this was done because we were interested in investigating whether different target ages would be differently affected by the flankers (due to different ages being either overestimated or underestimated when viewed alone, see Clifford, Watson, & White, 2018). However, we agree that the reviewer's suggested analysis is more likely to reveal subtle effects and have revised the manuscript to examine response error (perceived age - veridical age) averaged across the different target ages rather than raw response. When we do this, we replicate the effect of younger flankers, but we also now report a significant but smaller effect of older flankers. Please see the revised changes in the results section starting from page 12.

2) The construction of the ANOVAs is strange. You've got this factor Age Offsets which just doesn't make sense. As far as I can tell, it averages +15/-15 flankers, and +10/-10 flankers and so on. Now, a reasonable prior expectation is that the pluses and minuses should do opposite things to one another, and this this should be modulated by their magnitudes. Any such effect is basically cancelled out by the grouping within this factor. I think it needs more thought. My personal preference would be to rethink the whole analysis approach. I think you should ask much more targeted questions such as "is there a difference between +ve and -ve flankers?" and then design a statistic to address this. So, for example, you could just collapse across everything but flanker condition and compare these three (which of course, is exactly what the ANOVA does.) Then you could look at a plot of response error over flanker value – so running from -15 to +15 on the x-axis. Then you could look to see whether some reasonable curve could describe the data ... along with parameter confidence limits calculated through bootstrapping and so on. The whole approach would be much more targeted and would be much more in the vein of the sort of analysis that you find in the vision literature ... these are all techniques that your co-authors are very familiar with. At the moment, the whole analysis just comes across as a bit of a mess with a whole bunch of inscrutable interactions and various t-tests that are shoved into the supplementary materials and then largely ignored.

- As suggested, we re-analysed the data based on participants' response error rather than on their raw responses, averaged across target ages. This now allows us to also compare the size of the effects of the younger and older flankers in general rather than on each target age on its own. Based on this new analysis we now also report an effect of the older flankers (as explained above). This has also allowed us to substantially streamline our results for a more targeted analysis of the effects of the flankers on target age perception.

3) The age offset graphs are perhaps the most informative graphs in the paper. I don't understand why there are no error bars. Whilst we're on the subject, what do the error bars on the other graphs show? Standard errors? The t-test done on the age offset data – this can be done better. At the moment you're just doing (presumably) a t-test to look at the difference between two columns of 4 numbers each. So you're chucking away all that information about the variability within your participant pool. There are a bunch of approaches that you could take, your co-authors will have some ideas.

- The age offset graphs have been replaced by the response error graphs, since this presents our results in a clearer and more streamlined manner, and we now perform a two way ANOVA, using flankers condition (3 levels) and age offsets (4 levels) on the averaged response errors across the different target ages. We do not find an interaction between flankers' conditions and age offsets, therefore we compiled the response errors for the different flanked conditions together and ran separate t-tests on those. We have removed all analysis on the age offset data since there was no main effect of age offset. We apologise for the lack of clarity about the error bars. The figures now state that the error bars represent 95% confidence intervals (CI).

4) The idea that non-significance means no effect. This runs through the paper and is just plain wrong. If you want to make the assertion that there's no effect, then there are statistics that you can use – again, your co-authors will know. Otherwise, it needs to be expunged from the paper, which means changes in the abstract, results and discussion.

- Yes, this is true and we have changed this in the text. However now that we have performed the analysis on response error, we now find a significant (albeit smaller) effect of older flankers. In order to examine the magnitude of the contextual effect with younger flankers and older flankers we calculated (No flankers error – younger flankers' error) and contrasted it with (No flankers – Older flankers) thus comparing the absolute size of the response error magnitude (i.e. effect size) caused by the younger flankers and the older flankers. This analysis showed that the error caused by the younger flankers was significantly larger than the error caused by older flankers. This has now been discussed in the abstract, results and discussion of the revised manuscript.

5) The complexity of the third experiment. There are 6 factors – target age, target gender, flanker condition, age offset, flanker congruence and participant gender. It's really complicated ... to my mind, overcomplicated. I think, and this goes back to the points I've made above, a more targetted analysis would really help with this experiment. Also, from my reading you're using participant gender rather than gender congruence – why? I think you should be using the same approach as in the previous experiment. So the major issue I have with the paper is really with the entire conceptualisation of the analysis.

- Yes, it is true that there were many factors being tested/analysed in this third experiment. We have substantially compacted the data, looking at gender congruency and analysing the response error as in the previous experiments. We replicate our previous results from experiment I and II about the effects of flankers, and that it is not mediated by participant/flankers/targets gender congruency. Please see the revised changes on page 16 starting from line 358.

I also have a bunch of minor issues, these are as follows

1) P4 “perception of emotion” - personally, I think you perceive expression and then infer emotion. Others think different.

- Thank you, we have now revised the sentence to include “perception and deciphering of emotion”, which is how the original authors refer to it.

2) P5 – contrastive serial dependence with facial expressions. Have a look at the Liberman et al (2018) paper entitled “Serial dependence promotes the stability of perceived emotional expression depending on face similarity”.

- Thank you, this is indeed relevant to our study and we have included it in the revised introduction. However, please note that in the revised manuscript we have toned down the discussion of contrastive effects as it was mainly based on literature on

low level contextual effects rather than on high level facial attributes. We have reframed the hypothesis to state that we expect the assimilative effects to be modulated by the similarity between the target and the flankers. Please see the revised changes on page 6 lines 140-147.

3) You motivate your race/gender questions with the other race literature. I don't think it hangs together very well. Making people of different races make judgements of race is a bit different to people of different races or genders making judgements about age. Seeing how that might be differentially moderated by flanker is even more of a stretch. I think you need more/better motivation from the ingroup/outgroup literature.

- It is true that it's not well known how (or whether) ingroup or outgroup characteristics influence more complex judgements on secondary characteristics, such as the type of contextual effects we examine here. However, based on previously reported own-group biases in the perception of individual faces, we were interested in investigating whether such individual differences would also modulate contextual effects (even if these effects were not specifically on race but rather other facial characteristics). Other-Own race effects have been previously examined in temporal contextual effects. For example, Rhodes et al., (2009) found that other-race effect on face recognition was increased when participants were required to rate the attractiveness of own and other race faces. Another example is Kramer, Jones & Shirma (2013) who investigated sequential effects in judgements of facial attractiveness and found that when asked to judge the attractiveness of faces, participants first categorised the test faces by race then by gender which then modulated the attractiveness assimilation effects. Here we wanted to see whether similar effects were also present in non-temporally presented stimuli. However, we agree that further discussion would strengthen the manuscript and have now revised the manuscript to include a more detailed discussion of this. Please find the revised changes on page 6 starting from line 128.

4) P8 SONA – what is this?

- It's a research management system which is used to advertise studies and recruit participants for psychology experiments which is mainly used in the United Kingdom and Australia. The revised manuscript now clarifies what it is.

5) How did you set/maintain viewing distance?

- Viewing distance was maintained using a chin rest.

6) Pilot study. Really, I'd have liked to have seen the pilot study included in the main paper. I'm not a big fan of dumping stuff in supp mats. My view is that if you need it for the paper, then it should be in the paper. I can well understand that you might choose to keep the pilot in supp mats but can you please add a box plot of response minus target age over target age so that I can see something of the way the response distributions change over target age.

- Thank you for pointing this out. Since our new analysis has allowed us to streamline the paper, we have now included the pilot study in the paper instead of in the supplementary materials.

-

7) P13 bottom. “such that observers were tested on a single target-flanker age offset in a block”. Actually these appear to be negative/positive flanker pairs. To my mind, there are 8 age offsets, not 4.

- We agree technically there are 8 age offsets, but within each block in the experiments we only tested one absolute value of age offset (e.g. + 10 older flankers and -10 younger flankers). We have now clarified this on page X, line x-x.

8) Results ... I’ve commented in detail above about the results and I’d hope to see substantial changes and an increase in clarity with these ... but just to point out some of the non-sig means no effect errors – p19, sentence above Figure 5. P22, final sentence before General Discussion.

- Please see above for the changes in the analysis section.

9) P22. The first and penultimate sentences of the first paragraph basically say the same thing.

- Thank you, this has now been fixed.

10) P23. “this shift of perceived age of the target towards the flankers’ ages was only significant with younger flankers”. Firstly, I can’t remember seeing a statistic that actually supports this. I may have missed it, but I have been through the paper twice – so either it’s not there, or you’ve not done enough to draw it to your readers’ attention. Secondly, you can’t make anything of this unless you have a significant difference between the effect with younger flankers and the effect with older flankers. In other words, if A is significant and B is not, this does not necessarily imply a significant difference between A and B. Have a read of Nieuwenhuis et al (2011) *Erroneous analyses of interactions in neuroscience: a problem of significance. Nature Neuroscience, 14(9), 1105-1107.*

- This analysis has now changed since we analyse response error. Doing this we replicate the previous findings regarding the assimilative effects of younger flankers (i.e. target is perceived as younger when viewed with younger flankers than when viewed on its own), but we also find a similar (albeit smaller) effect of older flankers as well. To further compare the effects of younger and older flankers we compared the error magnitude caused by the two conditions and we find them to be significantly different with younger flankers causing a bigger error. This can be found in Figures 4, 5, and 7 for experiments 1, 2 and 3 respectively.